# The REGγ inhibitor NIP30 increases sensitivity to chemotherapy in p53-deficient tumor cells

Xiao Gao[1,2,3,14], Qingwei Wang[4,14], Ying Wang[5,14], Jiang Liu[5,14], Shuang Liu[6,14], Jian Liu [7], Xingli Zhou[3], Li Zhou[3], Hui Chen[3], Linian Pan[3], Jiwei Chen[3], Da Wang[1,2], Qing Zhang[6], Shihui Shen[3], Yu Xiao[3], Zhipeng Wu[1,2], Yiyun Cheng[3], Geng Chen [3], Syeda Krubra[3], Jun Qin[8], Lan Huang [9], Pei Zhang[10], Chuangui Wang[11], Robb E. Moses[12], David M. Lonard[12], Bert W. O' Malley [12], Fuad Fares[13], Bianhong Zhang[3✉], Xiaotao Li[1,12✉], Lei Li [3✉] & Jianru Xiao[1,2✉]

A major challenge in chemotherapy is chemotherapy resistance in cells lacking p53. Here we demonstrate that NIP30, an inhibitor of the oncogenic REGγ-proteasome, attenuates cancer cell growth and sensitizes p53-compromised cells to chemotherapeutic agents. NIP30 acts by binding to REGγ via an evolutionarily-conserved serine-rich domain with 4-serine phosphorylation. We find the cyclin-dependent phosphatase CDC25A is a key regulator for NIP30 phosphorylation and modulation of REGγ activity during the cell cycle or after DNA damage. We validate CDC25A-NIP30-REGγ mediated regulation of the REGγ target protein p21 in vivo using p53−/− and p53/REGγ double-deficient mice. Moreover, Phosphor-NIP30 mimetics significantly increase the growth inhibitory effect of chemotherapeutic agents in vitro and in vivo. Given that NIP30 is frequently mutated in the TCGA cancer database, our results provide insight into the regulatory pathway controlling the REGγ-proteasome in carcinogenesis and offer a novel approach to drug-resistant cancer therapy.

---

[1] East China Normal University and Shanghai Changzheng Hospital Joint Research Center for Orthopedic Oncology, East China Normal University, 500 Dongchuan Road, 200241 Shanghai, China. [2] Department of Orthopedic Oncology, Changzheng Hospital, The Second Military Medical University, 415 Fengyang Road, 200003 Shanghai, China. [3] Shanghai Key Laboratory of Regulatory Biology, Institute of Biomedical Sciences, School of Life Sciences, East China Normal University, 500 Dongchuan Road, 200241 Shanghai, China. [4] Department of Surgery, Department of Physiology & Cell Biology, College of Medicine, Davis Heart and Lung Research Institute, Wexner Medical Center, The Ohio State University, Columbus, OH 43210, USA. [5] The Institute of Aging Research, School of Medicine, Hangzhou Normal University, 310036 Hangzhou, Zhejiang, China. [6] Department of Hematology, Guangdong Second Provincial General Hospital, Guangzhou, Guangdong Province, P. R. China. [7] Reproductive & Developmental Biology Laboratory, National Institute of Environmental Health Sciences (NIEHS), Research Triangle Prk, NC 27709, USA. [8] The Joint Laboratory of Translational Medicine, National Center for Protein Sciences (Beijing) and Peking University Cancer Hospital, State Key Laboratory of Proteomics, Institute of Lifeomics, 102206 Beijing, China. [9] Department of Physiology and Biophysics, University of California, Irvine, CA 92697, USA. [10] Department of Pathology, The Second Chengdu Municipal Hospital, 610017 Chengdu, China. [11] Institute of Translational Medicine, Shanghai General Hospital, Shanghai Jiao Tong University School of Medicine, Shanghai, China. [12] Department of Molecular and Cellular Biology, Dan L. Duncan Cancer Center, Baylor College of Medicine, One Baylor Plaza, Houston, TX 77030, USA. [13] Department of Human Biology. Faculty of Natural Sciences, University of Haifa, Haifa 3498838, Israel. [14] These authors contributed equally: Xiao Gao, Qingwei Wang, Ying Wang, Jiang Liu, Shuang Liu. ✉email: zhkyzhbh.549803@163.com; xiaotaol@bcm.edu; lllkzj@163.com; jianruxiao83@163.com

Ubiquitin-independent proteasome degradation of proteins has emerged as a field with promising translational potential[1]. The proteasome activator REGγ (also known as PA28γ, PAME3, Ki antigen) belongs to the REG or 11S family of proteasome activators that can bind and activate the 20S proteasome in the absence of ubiquitin and ATP[2], distinct from the canonical ubiquitin–proteasome pathway[3]. Since the discovery of the transcriptional coregulator, SRC-3, as a substrate, REGγ has been demonstrated to mediate an alternate means of digesting multiple proteins, including the cell-cycle regulator p21[1,4]. REGγ-null mouse embryonic fibroblasts display lower saturation density than wild-type counterparts, and a decreased proportion of S-phase cells[5]. Evidence also suggests that REGγ plays an important role in carcinogenesis[6,7], where regulation of the tumor suppressor p21 appears to be critical. However, mechanisms controlling the oncogenic REGγ proteasome pathway remain unknown.

Progression through the cell cycle is controlled by the induction of cyclins and activation of cognate cyclin-dependent kinases. Key transitions in the cell cycle are regulated by a family of serine/threonine protein kinases termed cyclin-dependent kinases (CDKs). CDKs are regulated by multiple factors associated with members of heterologous small regulatory proteins, such as p16 and p21[8]. As a universal inhibitor of cyclin-CDK complexes[9–11], p21 is activated by tumor suppressor p53 to inhibit Cdk2/cyclin E complexes, leading to arrest of cells at the G1/S checkpoint[12], as well as G2/M-phase arrest[13]. It is generally accepted that p21 can be degraded by both ubiquitin-dependent and ubiquitin-independent proteasome pathways[1,14,15]. Three E3 ubiquitin ligase complexes, SCFSKP2 (SKP1-CUL1-SKP2), CRL4CDT2 (CUL4A or CUL4B-DDB1-CDT2), and APC/CCDC20 (anaphase-promoting complex (APC-cell division cycle 20) promote the ubiquitylation and degradation of p21[14]. After mutating all six lysine residues in p21 (the sites of potential ubiquitin conjugation) to arginine (p21K6R) to prevent poly-ubiquitination, p21 can still be degraded by the REGγ proteasome[15], as shown by us and others[1,16]. Despite a decrease of p21 stability following DNA damage[17–20], the contribution of the ubiquitin-independent proteasome system to p21 degradation after DNA damage remains unclear. In addition, whether p21 can be degraded during a specific stage of the cell cycle by the ubiquitin-independent proteasome pathway is still under debate.

CDC25A acts in cell-cycle regulation by activating CDKs to allow for cell-cycle progression. The removal of inhibitory phosphorylation on CDK proteins by dual-specificity phosphatases of the CDC25 family is a crucial step in the activation of CDK-cyclin complexes. The phosphatase CDC25 has three known isoforms in mammals: CDC25A, CDC25B, and CDC25C[21]. Mouse knockout models have revealed that double-knockout CDC25B–CDC25C mice develop normally and cells from these mice display normal cell-cycle profiles[22]. However, a CDC25A knockout is lethal at a very early-stage embryogenesis, indicating that CDC25A is required for critical functions during cell division[23]. CDC25A is a short-lived protein that is specifically degraded in response to DNA damage[24]; phosphorylation of CDC25A triggers ubiquitin-mediated degradation of checkpoint kinases in response to UV exposure[25,26]. Checkpoint kinase 1 (Chk1) and serine–threonine kinase38 (STK) can phosphorylate Ser-76 in CDC25A following DNA damage, thereby promoting its degradation[24,27]. Though p21 was reported to regulate CDC25A[28], it is not known whether CDC25A can affect p21 stability independently of p53 during the cell cycle.

NEFA-interacting nuclear protein 30 (NIP30 also known as PIP30) was reported in a BioGRID database analysis[29] and is expressed in skeletal muscle[30]. NIP30 conserves an N-terminal bipartite nuclear localization signal (NLS) in its nuclear localization domain and has been demonstrated in nuclei of the C2C12 cell line[30]. The C-terminus of NIP30 determines protein binding to REGγ and affects REGγ-20S proteasome peptide-degradation function, leading to negative control of REGγ in Cajal bodies[31]. Jonik-Nowak et al. demonstrated that NIP30 affects REGγ interactions with cellular proteins, including the 20S proteasome. NIP30 is an important regulator of REGγ in cells, and controls the multiple functions of the proteasome within the nucleus[31]. However, whether NIP30 functions as a tumor suppressor, and how upstream signals regulate NIP30 is unknown. Here, we report that the REGγ-20S proteasome pathway is regulated by NIP30, a REGγ inhibitor. We demonstrate that NIP30 attenuates decay of REGγ target proteins, including p21, and inhibits tumor cell growth via association with REGγ and blockage of REGγ-dependent proteasome activation. As an upstream regulator, CDC25A functions to antagonize NIP30 and promote REGγ-directed p21 degradation, independent of p53 during the cell cycle or after DNA damage. Phosphorylation-mimetic-NIP30 cancer cells show enhanced sensitivity to chemotherapeutic agents in vitro or in vivo. Our results suggest a mechanism for regulation of the ubiquitin-independent REGγ-proteasome system that is crucial for modulation of carcinogenesis.

## Results

**NIP30 is a REGγ inhibitor**. To search for REGγ regulatory proteins, we carried out mass spectrometric analyses (Fig. 1a) as well as yeast two-hybrid assays (Fig. 1b) to screen for REGγ-interacting proteins. Results from both experimental approaches identified NIP30 as a REGγ-associated protein. We validated the interactions between REGγ and NIP30 by reciprocal co-immunoprecipitation (co-IP) and by repeated yeast two-hybrid testing (Fig. 1c; Supplementary Fig. 1a). Manipulation of REGγ by transient overexpression (Supplementary Fig. 1b) or depletion had no effect on the expression of NIP30 (Supplementary Fig. 1b, c, d), suggesting that NIP30 is not a target of the REGγ proteasome. Transient overexpression of NIP30 in 293T or H1299 cells (both having reduced p53 expression) significantly enhanced p21 protein levels (Fig. 1d; Supplementary Fig. 2a), but not p21 mRNA expression (Supplementary Fig. 2b). P21 expression also was not altered in cells (293T KO) knocked out for REGγ (Fig. 1d). In contrast, knocking down NIP30 resulted in a significant reduction in p21 protein level (Fig. 1e; Supplementary Fig. 2c, d), suggesting a role for NIP30 in blocking REGγ-mediated degradation of p21.

To substantiate whether NIP30 really inhibits REGγ activity, we utilized a 293-cell system, in which overexpression of a wild-type (WT) REGγ or an enzymatically inactive protein (N151Y) can be induced[4]. The results showed that overexpression of NIP30 in the 293-cell system blocked the degradation of not only p21 (Supplementary Fig. 2f) but also HCV (Fig. 1f) and Lats1 (Supplementary Fig. 2g), known substrates of the REGγ proteasome[6], only when wild-type REGγ, but not the inactive (N151Y) REGγ, was expressed. Silencing or overexpressing NIP30 in cells with or without REGγ had no effect on mRNA levels of p21 (Supplementary Fig. 2b, e), suggesting a post-transcriptional regulation of p21 by NIP30 via REGγ. These data indicate that NIP30 acts as an inhibitor of REGγ-mediated degradation of substrate proteins.

**NIP30 directly interacts with REGγ**. We found REGγ mediates degradation of p21 in a p53-independent manner[1,6]. To define the mechanism of inhibiton of degradation, we focused on p21 as a readout, using cells with inactive p53 to exclude influences by p53. We generated a series of constructs with deletions to determine the REGγ-interactive domain(s) in NIP30. Co-IP and

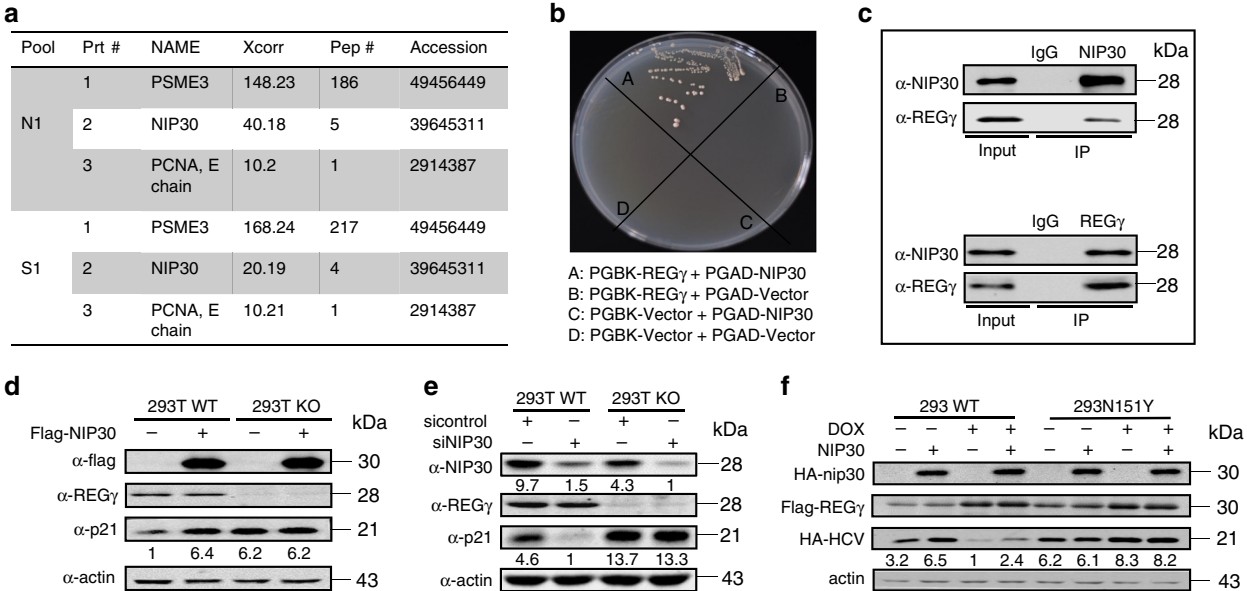

**Fig. 1 NIP30 interacts with REGγ and antagonizes REGγ-mediated degradation of p21. a** Mass spectrum analysis of proteins co-immunoprecipitated with Flag-REGγ stably expressed in 293T cells. N1 stands for a nuclear fraction; S1 refers to a cytoplasmic fraction. **b** Yeast two-hybrid assays demonstrated physical interactions between NIP30 and REGγ. **c** Intracellular interactions between REGγ and NIP30 in 293T cells determined by reciprocal co-immunoprecipitation and western blot analysis. **d** Overexpression of Flag-NIP30 in 293T WT, but not in REGγ KO cells, induced elevation of p21. **e** Silencing NIP30 for 72 h significantly reduced expression of p21 in 293T cells with normal REGγ. **f** NIP30 inhibited REGγ-dependent degradation of transiently expressed non-mammalian substrate, hepatitis virus C core protein (HCV), in HEK293 cells inducibly expressing WT REGγ, but not in cells with an inactive form (N151Y). All data in this figure are representative of three independent repeats. Source data are provided in a Source Data file.

interaction domain analysis (Fig. 2a; Supplementary Fig. 3a) showed that the region including amino acids 219–231 in NIP30 was critical for its interaction with REGγ (Fig. 2b). Sequence alignment indicates that the REGγ-interacting motif in NIP30 is an evolutionarily conserved serine-rich domain (Fig. 2c), suggesting a potential site for posttranslational modification. To determine whether posttranslational modification of the serine residues is essential for NIP30 function, as reported[31], we mutated each of the serine residues within the 219–231 region to alanine (A) or to aspartic acid (D) to produce phosphor-defective and phosphor-mimetic NIP30, respectively. Individual mutations in NIP30 residues 226–230 resulted in partially attenuated binding with REGγ (Supplementary Fig. 3b), suggesting contribution of all the four serine sites. This idea was tested by using a combination of multiple mutations in three modules: 221–224 (for the serines at 221, 222, and 224), 226–230 (for the serines at 226, 227, 228, and 230), and 221–230 (for all the seven serines in this region), namely S221–224A, S221–224D, S226–230A, S226–230D, S221–230A, and S221–230D, in line with the report that a phosphorylated C-terminus of NIP30 is required for the REGγ interaction[31]. Two of these modules containing the four mutations of S residues to A in 226–230 (abbreviated as "4A" hereafter) disrupted interactions with REGγ, while the S226–230D (abbreviated as "4D" hereafter). The S221–230D mutations preserved the interaction (Fig. 2d), indicating an important role for phosphorylation in the regulation of the NIP30-REGγ complex formation. To substantiate the requirement for phosphorylation, transiently expressed NIP30 wild-type (WT) and the 4A construct were immunoprecipitated for phosphatase treatment. The λPP-treated WT-NIP30 migrated faster, while λPP-treatment of the 4A did not increase migration rate (Supplementary Fig. 3c), indicating that NIP30 is phosphorylated in the 226–230 region. Intracellular phosphorylation of NIP30 was verified by LC–MS/MS analysis of immunoprecipitated NIP30 (Supplementary Fig. 3d, e, Supplementary Tables 3 and 4).

To understand the role of phosphorylation/dephosphorylation in NIP30-mediated regulation of REGγ functions, we analyzed the impact of NIP30 on the activity of REGγ in cell-free proteolysis using purified bacterially expressed NIP30-WT, NIP30 4A, NIP30-4D proteins and in vitro translated p21 from rabbit reticulocyte system (TNT) (Supplementary Fig. 3f–i). Only NIP30 4D, but not NIP30 4A or the NIP30 WT that cannot be phosphorylated in bacteria, significantly inhibited REGγ proteasome-driven degradation of p21 (Supplementary Fig. 3j, k). p21 alone, along with p21 + 20S and p21 + REGγ (the first three lanes in Supplementary Fig. 3j), were used as controls to demonstrate that p21 does not self-degrade, and latent (inactivated) 20S proteasome or REGγ alone cannot degrade p21. We then generated cell lines stably expressing WT-NIP30, phosphorylation-defective NIP30 4A (S226–230A) or phosphorylation-mimetic NIP30-4D (S226–230D) mutants in 293T and H1299 cells to evaluate NIP30-mediated regulation of p21 on cell growth. Cells overexpressing NIP30 WT or the phosphorylation-mimetic NIP30-4D construct resulted in elevated p21 compared with control cells, but there was no significant increase of p21 in NIP30-4A expressing cells or in cells depleted of REGγ (Fig. 2e; Supplementary Fig. 3l). Viability assays indicated that the NIP30-4D construct led to a dramatic inhibition of cell growth whereas the NIP30-4A construct had minimal inhibition. None of the NIP30 constructs had significant impact on cell growth in 293T-REGγ KO cells (Fig. 2f; Supplementary Fig. 3m), indicating a REGγ-dependent action of NIP30. Xenograft tumors generated from these cells established that phosphorylation is a determinant for the tumor suppressive role of NIP30 in REGγ-dependent cell growth. Tumor growth was retarded by overexpressed NIP30 WT or NIP30 4D, but not the functionally defective NIP30 4A (Fig. 2g; Supplementary Fig. 3n). These results demonstrate that targeted phosphorylation within residues 226 to 230 of NIP30 dictates REGγ proteasome function.

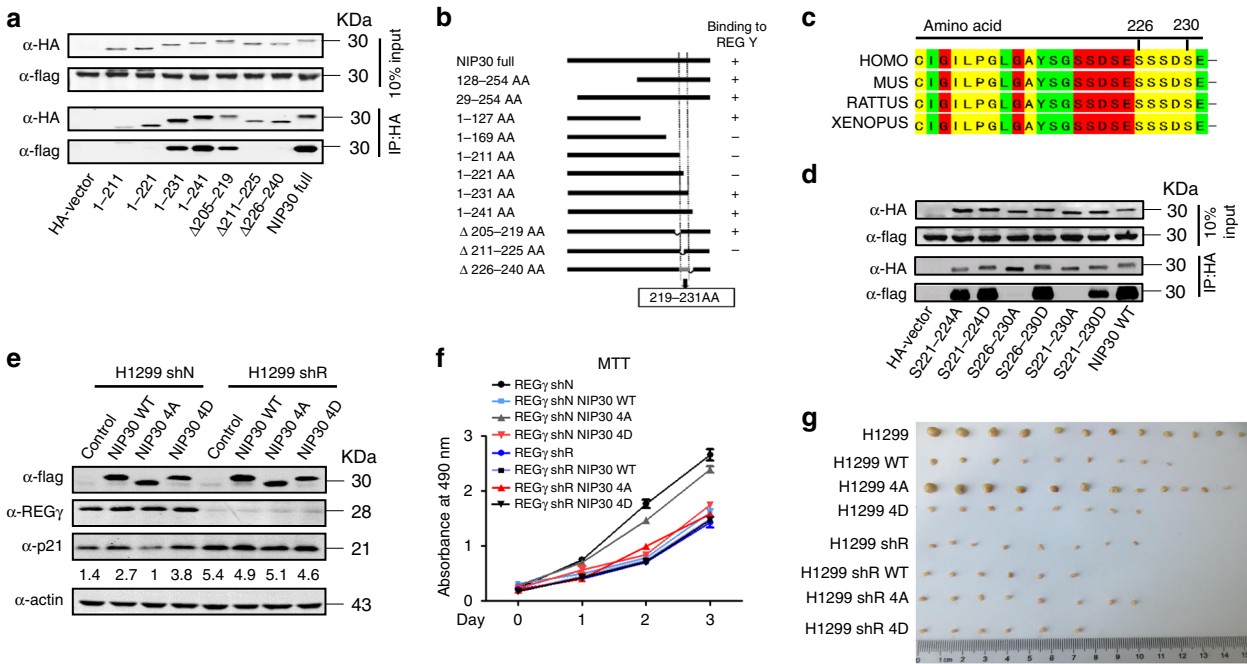

**Fig. 2 NIP30 regulates REGγ through phosphorylation of its conserved binding motif. a** A panel of NIP30 truncations co-expressed with flag-REGγ in 293T cells was analyzed for interaction domains by immunoprecipitation. **b** A summary of the interaction between REGγ and various NIP30 derivative clones (+ for binding, − for no binding). **c** Multiple sequence alignment showing the evolutionarily conserved, serine-rich REGγ-interacting motif in NIP30. **d** NIP30 is phosphorylated at 4 specific serine residues. HA-tagged NIP30 with phosphorylation-mimetic (S/D) or defective (S/A) mutations were tested for interaction with REGγ by immunoprecipitation. **e** Generation of NIP30 expressing cells. H1299 cells with or without REGγ (shN/shR) stably expressing NIP30 constructs (WT, 4A, or 4D) were examined for p21 expression. **f** H1299 cells expressing distinct NIP30 constructs were analyzed for cell proliferation by MTT assays. Data represent mean ± SEM (n = 4). **g** Xenograft tumors derived from H1299 cells stably expressing different NIP30 constructs (WT/4A/4D) were compared for tumor size. Quantitative results are in Supplementary Fig. 3n. Data in this figure are representative of three independent repeats. Source data are provided in a Source Data file.

**CDC25A is a regulator for NIP30 phosphorylation.** Identification of molecules modulating NIP30 phosphorylation might help determine if NIP30 is a hub of signaling, leading to regulation of the REGγ proteasome activity. Using phosphor-specific antibodies against S288 and S230 of NIP30 (Supplementary Fig. 4a, b), we screened multiple upstream candidate kinases including Casein Kinase 2 (CK2/CSNK2), p38, Erk1/2, PI₃K, and JNK that might phosphorylate NIP30 based on consensus kinase-binding sequences around the NIP30 226–230 motif (Supplementary Fig. 5a). CK2 catalytic subunits (CK2α or CK2α') could bind with NIP30 (Supplementary Fig. 5b) by co-immunoprecipitation assays. Overexpression of CK2α or CK2α' led to increased phosphorylation of NIP30 (Supplementary Fig. 5c, d), consistent with the report that the NIP30 C-terminus can be phosphorylated by CKII[31]. However, inhibitors to kinases, including p38, JNK, CK1, and GSK3β, could reduce phosphorylation in NIP30 and attenuate NIP30-REGγ interaction (data not shown), indicating that NIP30 phosphorylation might result from action of multiple kinases. Therefore, we wondered if the regulatory mechanism for rapid alteration of NIP30 phosphorylation might result from specific phosphatase action. We screened a human Ser/Thr phosphatase library[32] (Supplementary Fig. 5e), in which CDC25A, a phosphatase essential for G1-S/G2-M transition and activation of the cell-cycle kinase cyclin E-cdk2, was found to eliminate phosphorylation of NIP30 at 228 and 230 and therefore to potentially regulate REGγ degradation activity on targets such as p21 (Supplementary Fig. 5f–g; Fig. 3a).

To determine if CDC25A regulates the NIP30-REGγ-p21 axis, CDC25A was overexpressed or silenced in 239T cells with or without REGγ. Although CDC25A limited phosphorylation of NIP30 in both 293T WT and 293T KO cells, reduction in p21

only occurred in 293T WT cells overexpressing CDC25A (Fig. 3b). In contrast, depletion of CDC25A resulted in elevation of NIP30 phosphorylation and p21 levels in a REGγ-dependent manner (Fig. 3c). Using a CDC25A inhibitor (NSC95397) for manipulation of CDC25A levels, we found that CDC25A controls REGγ-dependent degradation of a panel of substrate proteins, including p21, p16, p19, and GSK3β, by altering phosphorylation of NIP30 (Fig. 3d; Supplementary Fig. 5h, i, j, k). We verified that CDC25A directly dephosphorylates NIP30 by in vitro incubation of bacterially purified CDC25A or immunoprecipitated cellular CDC25A with p-NIP30 proteins for various times followed by Western blot analysis of NIP30 phosphorylation status (Fig. 3e; Supplementary Fig. 5l). Physical interaction between NIP30 and CDC25A was verified by reciprocal co-IP experiments (Fig. 3f). These results demonstrate that CDC25A is a key effector in control of NIP30 phosphorylation levels.

**The CDC25A–NIP30 pathway regulates p21 during cell cycle.** CDC25A is a rapidly turned over protein with levels regulated by periodic synthesis and ubiquitin-mediated proteolysis[33]. Since CDC25A activity varies in specific cell-cycle stages[34–36], we analyzed expression of CDC25A in p53-compromised cells synchronized in G1, S, or G2 phases (Supplementary Fig. 6a, b). Regardless of REGγ levels, CDC25A was enriched in 293T, and H1299 cells synchronized in S phase (Fig. 4a; Supplementary Fig. 6c). Increased activity of CDC25A in the S phase was correlated with concomitant dephosphorylation of NIP30 and an increased degradation of p21 in a REGγ-dependent manner (Fig. 4a; Supplementary Fig. 7c). A CDC25A inhibitor, NSC95397, abolished dephosphorylation of NIP30 and prevented

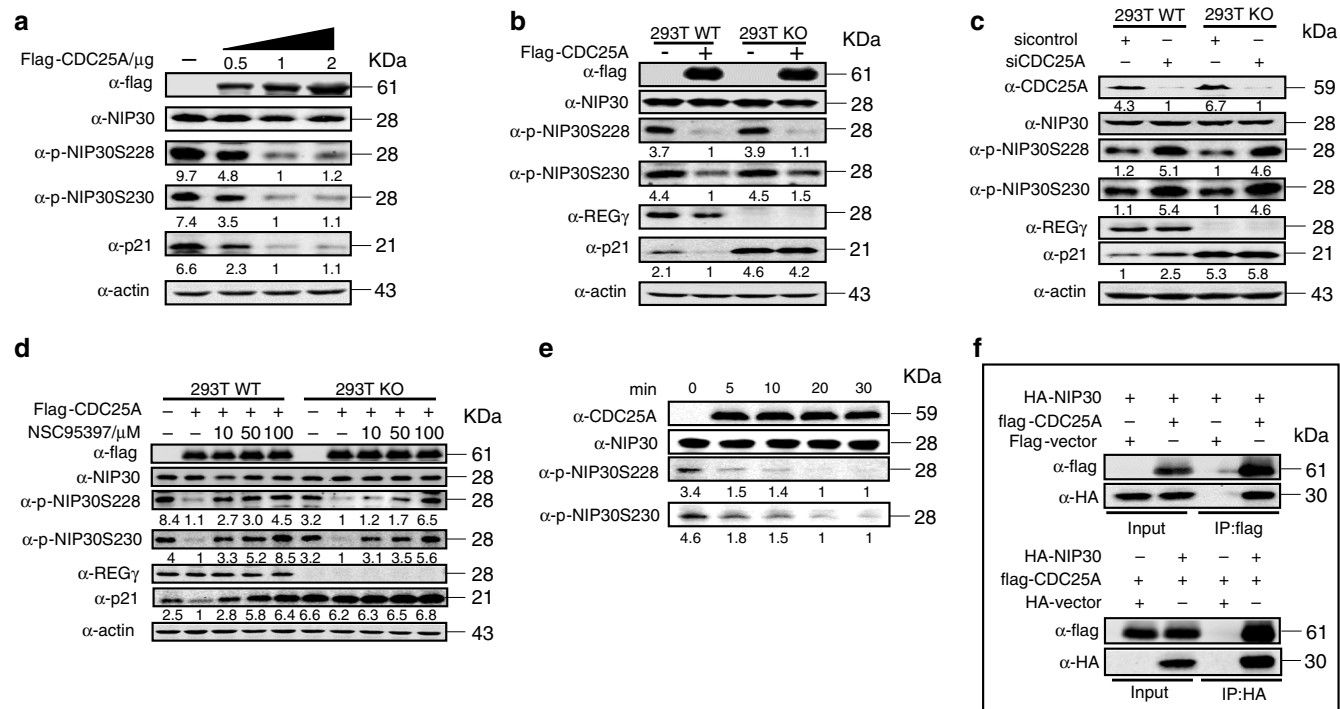

**Fig. 3 NIP30 pS228/pS230 is dephosphorylated by CDC25A. a** 293T cells were transiently transfected with 0.5 μg, 1 μg, or 2 μg of flag-CDC25A plasmid for 48 h, and the total cell lysates were analyzed by immunoblotting with antibodies against flag, total NIP30, pNIP30Ser228, pNIP30Ser230, p21, and actin. Data are representative of three independent experiments. **b** REGγ WT and knockout 293T cells were transiently transfected with 2 μg flag-CDC25A plasmid for 48 h, and the total cell lysates were analyzed by immunoblotting with indicated antibodies. Data represent mean ± SEM from three independent experiments. **c** REGγ WT and 293T KO cells were transfected with indicated short interfering RNA (-si) against CDC25A or irrelevant siRNA (si control) for 72 h. Cell lysates were subjected to western blot analysis with indicated antibodies. Data represent mean ± SEM from three independent experiments. **d** REGγ WT and 293T KO cells were transiently transfected with 2 μg of flag-CDC25A plasmid for 48 h and then treated with NSC95397 10 μM, 50 μM, 100 μM for 3 h before harvest. Cell lysates were subjected to western blot analysis using indicated antibodies in three independent experiments. Data represent mean ± SEM. **e** Immunoprecipitated cellular NIP30 was incubated with bacterially purified CDC25A protein for indicated times. The lane labeled "0" was also incubated for 30 min at 30 °C. Data represent mean ± SEM from three independent experiments. **f** 293T cells were transfected with HA-NIP30 and flag-CDC25A or a flag-control vector for 36 h. Reciprocal interactions between NIP30 and CDC25A were analyzed by immunoprecipitation assays with Flag-M2 agarose beads or HA beads. Data represent mean ± SEM from three independent experiments. Source data are provided in a Source Data file.

REGγ-mediated degradation of p21 during the S phase (Fig. 4b). To exclude potential off-target effects of the CDC25A inhibitor, we employed RNAi, to eliminate CDC25A function. Silencing CDC25A resulted in accumulation of phosphor-NIP30 and p21 protein levels equivalent to that by the CDC25A inhibitor (Fig. 4c), while CDC25A overexpression augmented REGγ-dependent degradation of p21 (Fig. 4c).

To ensure CDC25A regulates REGγ function via NIP30, we silenced NIP30 in the presence or absence of the CDC25A inhibitor. Knockdown of NIP30 effectively prevented restoration of p21 levels in cells treated with the CDC25A inhibitor (Fig. 4d), indicating that NIP30 acts downstream of CDC25A. Furthermore, overexpression of phosphorylation-mimetic NIP30 4D, capable of binding and inhibiting REGγ activity, allowed accumulation of more p21 than the phosphorylation-defective NIP30-4A mutant (Fig. 4e). Consequently, promoting NIP30 or reducing CDC25A activities in cells led to cell-cycle arrest in G1 or vice versa (Supplementary Fig. 6d, e, f). Taken together, we conclude that the CDC25A–NIP30–REGγ-regulatory pathway significantly contributes to regulation of p21 during the cell cycle.

**The CDC25A–NIP30 pathway controls p21 levels following DNA damage.** Given that CDC25A is rapidly degraded upon genotoxic stress[24,27], we investigated p53-independent regulation of p21 by the CDC25A–NIP30-REGγ pathway following DNA damage. When p53-deficient cells were treated with UV irradiation at indicated doses, a sharp decrease in CDC25A was observed with concurrent increase in phosphorylation of NIP30 at S228/230 and p21 accumulation (Fig. 5a–c; Supplementary Fig. 7a). Since UV-induced accumulation of p21 protein occurred only in REGγ-positive, but not in REGγ-deficient cells, and no changes were noted in p21 mRNA levels before or after DNA damage (Supplementary Fig. 7b, c), this response is REGγ-dependent. To define the regulatory hierarchy in the CDC25A–NIP30-REGγ pathway following DNA damage, we silenced CDC25A or NIP30, respectively, prior to UV irradiation. While depletion of CDC25A led to increase of NIP30 phosphorylation and p21 accumulation after UV irradiation (Fig. 5b; Supplementary Fig. 7d), NIP30 knockdown significantly blocked UV-induced accumulation of p21 in REGγ WT cells (Fig. 5c; Supplementary Fig. 7e), suggesting that CDC25A acts upstream of NIP30.

To ensure DNA damage other than UV irradiation can also induce p21 accumulation via the CDC25A–NIP30-REGγ pathway, we exposed cells to methanesulfonate (MMS) or doxorubicin (Dox), chemical reagents that can trigger the DNA damage response. Treatment with MMS (Fig. 5d; Supplementary Fig. 7f) or Dox (Fig. 5e) led to dose-dependent effects on degradation of CDC25A, NIP30 phosphorylation, and p21 accumulation similar to those after UV irradiation, without changes in p21 mRNA (Supplementary Fig. 7g, h). Silencing CDC25A or NIP30 in 293T

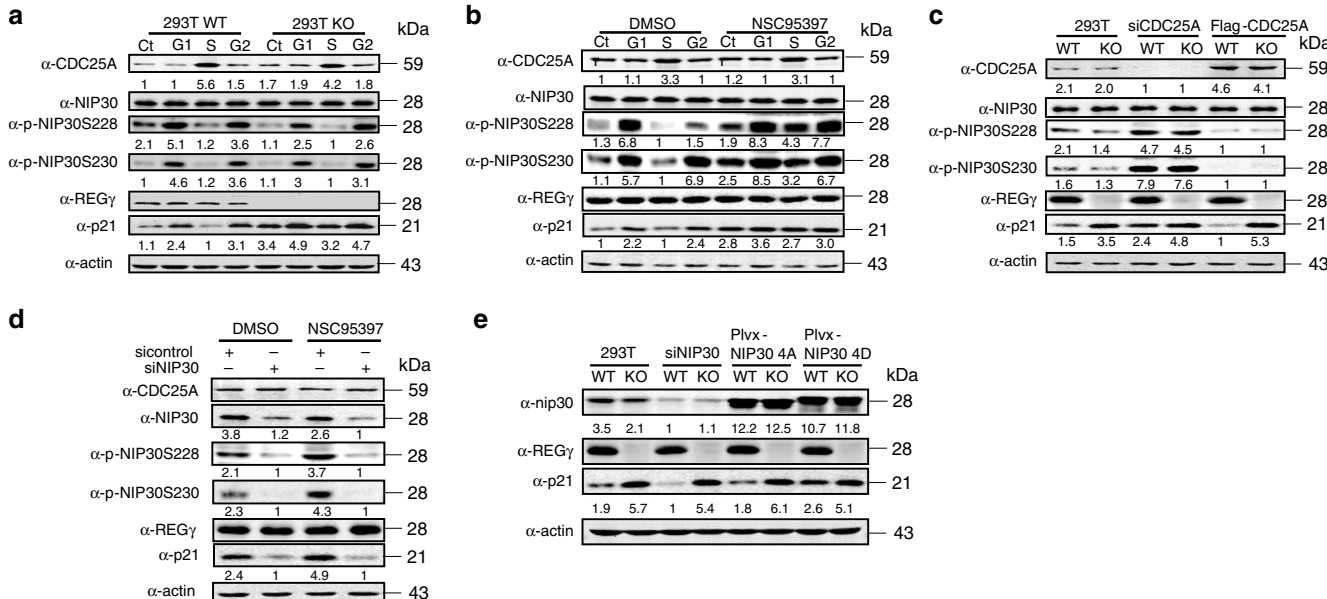

**Fig. 4 The CDC25A–NIP30-REGγ pathway regulates p21 during the cell cycle. a**, REGγ WT and 293T KO cells were synchronized to the G0/G1, S, and G2/M phases. Cell lysates were subjected to western blot analysis with indicated antibodies. Data are representative of at least three independent experiments. **b** Synchronized REGγ WT and KO 293T cells were treated with NSC95397 for 3 h. Cell lysates were subjected to western blot analysis. Data represent mean ± SEM from three independent experiments. **c** REGγ WT and KO 293T cells were transfected with an siRNA specific for CDC25A or flag-CDC25A for 72 h. Cells synchronized in the S phase were collected for western blot analysis. Data represent mean ± SEM from three independent experiments. **d** 293T cells were transfected with a short interfering RNA (-si) for NIP30 or an si control for 72 h. Cells arrested at the S phase were collected for western blot analysis in three independent experiments. Data represent mean ± SEM. **e** REGγ WT and KO 293T cells were transfected with indicated short interfering RNA (-si) for NIP30 or NIP30 4A/4D constructs. Cells synchronized at the S phase were collected for western blots in three independent experiments. Data represent mean ± SEM. Source data are provided in a Source Data file.

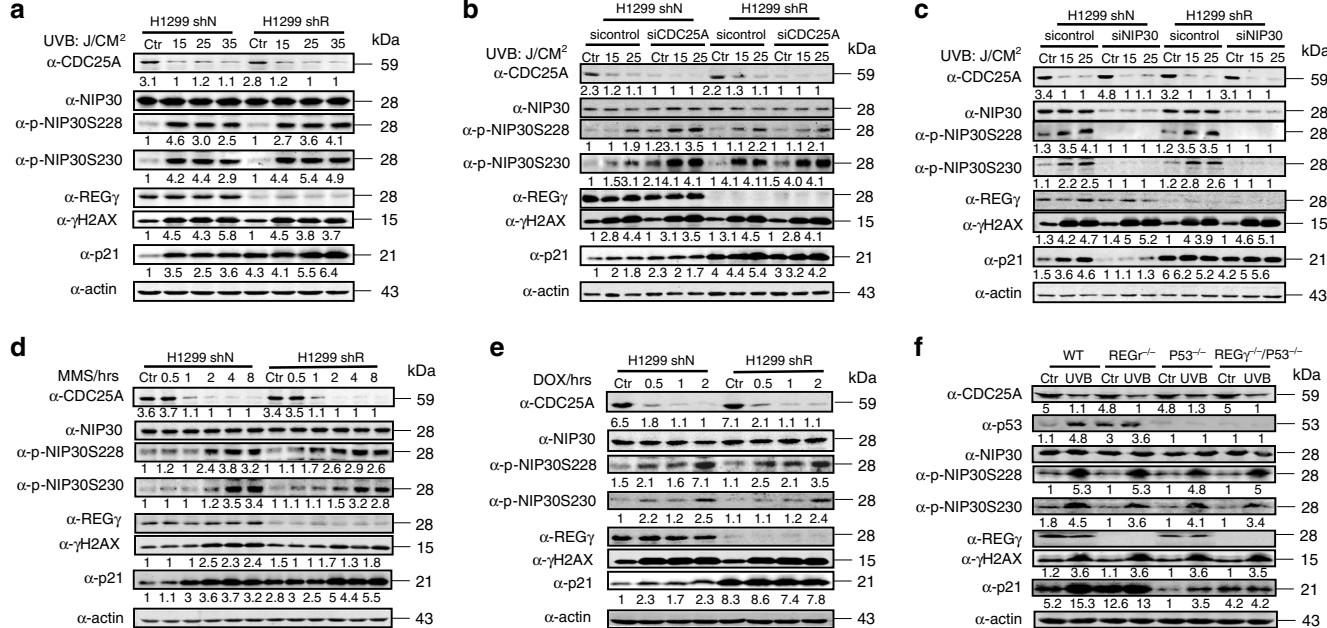

**Fig. 5 The CDC25A–NIP30-REGγ pathway controls p21 following DNA damage. a** H1299 REGγ shN and shR cells were irradiated with 0, 15, 25, or 35 J/m² UVB and harvested 12 h later. Cell lysates were analyzed by western blots with indicated antibodies. **b** H1299 REGγ shN and shR cells were treated with siRNA against CDC25A or a control siRNA followed by UVB irradiation. Cell lysates were subjected to western blot with indicated antibodies. **c** H1299 REGγ shN and shR cells were treated with siRNA against NIP30 or a control siRNA followed by UVB irradiation. Cell lysates were subjected to western blot with indicated antibodies. **d**, **e** CDC25A–NIP30 action after DNA damage by chemical reagents. H1299 REGγ shN and shR cells were treated with 0.2 mM MMS (**d**) or 10 μg/ml Doxorubicin (**e**) for the indicated time course. The levels of CDC25A, total NIP30, pNIP30Ser228, pNIP30Ser230, p21, REGγ, and γH2AX were determined by western blot. **f** In vivo action of the CDC25A–NIP30-REGγ pathway. REGγ$^{+/+}$, REGγ$^{-/-}$, P53$^{-/-}$, and REGγ$^{-/-}$/P53$^{-/-}$ newborn mice were exposed to UV light as described in "Methods". Skin samples from exposed dorsal side or ventral side (un-exposed control) were examined for proteins indicated. All the experiments were repeated three times. Source data are provided in a Source Data file.

or H1299 cells followed by MMS treatment resulted in the same changes in p-NIP30 and p21 as seen after UV-induced damage (Supplementary Fig. 7i–l).

To address whether the CDC25A–NIP30-REGγ pathway has physiological functions independent of p53 in vivo, we generated REGγ[+/+], REGγ[−/−], P53[−/−], and REGγ[−/−]/P53[−/−] mice for UV irradiation studies. UV-mediated degradation of CDC25A led to accumulation of p-NIP30 in the skin of newborn WT, p53[−/−], REGγ[−/−], and p53[−/−]/REGγ[−/−] mice. UV-induced stabilization of p21 was observed in both WT and p53−/− skin, but no differences in p21 were noted before or after UV treatment for REGγ[−/−] and p53[−/−]/REGγ[−/−] double-knockout mice (Fig. 5f). These results suggest that the CDC25A–NIP30-REGγ pathway is important in p53-independent regulation of p21 following DNA damage in vitro and in vivo.

**Phosphorylation of NIP30 sensitizes chemotherapeutic action.** To verify putative tumor suppressive function of NIP30, we analyzed the TCGA database for a panel of human cancers. The results disclosed relatively high frequency of deletions, point mutations, and amplifications in the NIP30-encoding gene (Fig. 6a; Supplementary Fig. 8a, Supplementary Tables 1 and 2), implicating NIP30 in tumor development.

Lack of functional p53 renders cells resistant to the effect of multiple anticancer drugs[37], including 5-fluorouracil[38], doxorubicin[39], cisplatin[40]. Given the tumor suppressive role of p-NIP30 in cells with p53 dysfunction, we tested whether NIP30 4D may sensitize p53-deficient H1299 cells to various anticancer drugs in vitro and in vivo. H1299 cells and H1299 stably overexpressing

NIP30 4D showed enhanced growth inhibition by etoposide, 5-fluorouracil, cisplatin, and doxorubicin, whereas the NIP30-4A cells did not (Fig. 6b). Xenograft tumors generated from H1299 cells and H1299 stably overexpressing NIP30 4D, but not NIP30 4A, had strikingly reduced tumor volumes following treatment with cisplatin, doxorubicin, or 5-fluorouracil (Fig. 6c, d), suggesting that blockade of the REGγ pathway by its inhibitor, NIP30, dramatically increases anticancer sensitivity in p53-deficient tumor cells.

We propose a model in which accumulation or degradation of CDC25A during the cell cycle or after DNA damage leads to regulation of REGγ via NIP30, controlling p21 levels in a p53-independent manner (Fig. 6e). We do not know whether activated CDC25A acts on NIP30 in isolation or in a complex with other proteins. Overall, this study defines NIP30, a REGγ inhibitor, as a putative tumor suppressor, acting as a molecular switch for integration of upstream signals to modulate the REGγ proteasome function.

## Discussion

REGγ-dependent degradation of the cell-cycle regulator p21 is known[1,16], but the regulatory mechanism for REGγ action remained largely unclear. In this study, we have uncovered a mechanism in which the oncogenic REGγ pathway is negatively controlled by NIP30. This study demonstrates that NIP30 is a potential tumor suppressor and acts by blocking the degradation of REGγ substrates. Active NIP30 is phosphorylated, which allows binding to and inhibition of REGγ. The REGγ-NIP30 complex is subject to an upstream phosphatase, CDC25A, which

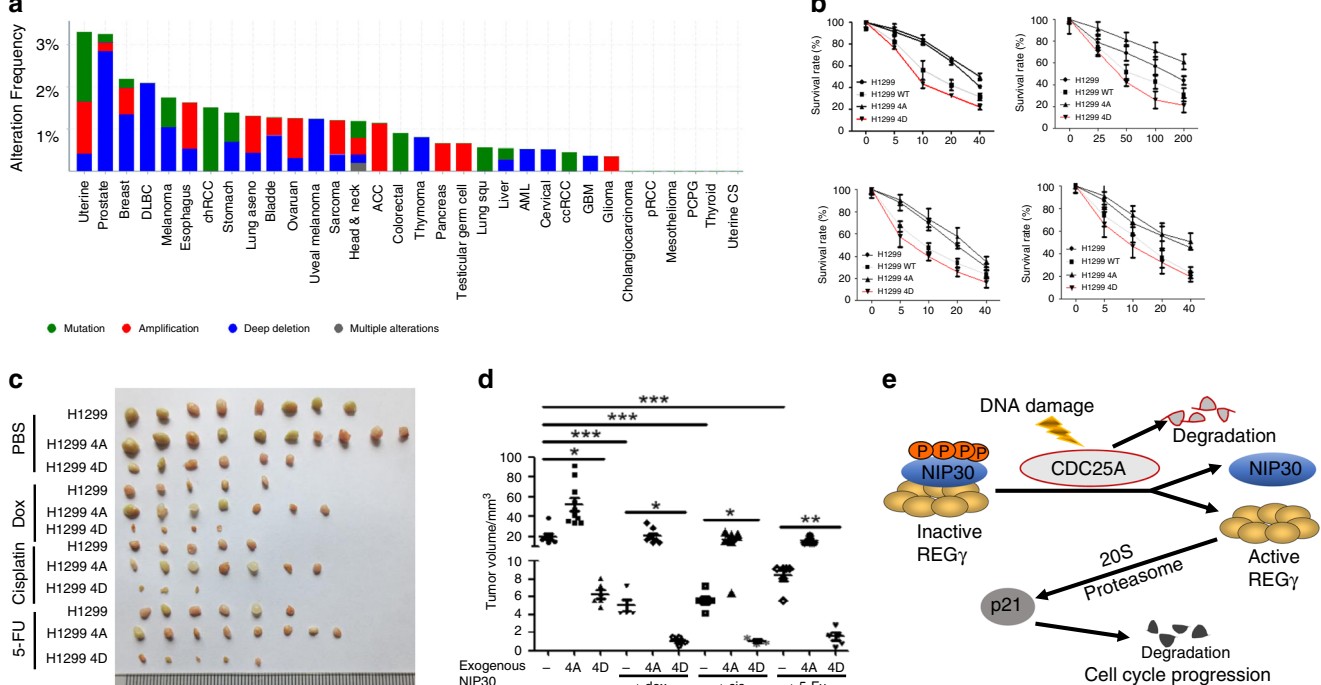

**Fig. 6 NIP30 mimetics enhance chemotherapeutic effects in p53-deficient cells. a** Summary of alterations for NIP30 in different cancer types from the TCGA project. The alteration types include mutation (green), amplification (red), deep deletion (blue) and multiple alternations (gray). **b** NIP30 phosphorylation mimetics enhance anticancer drug efficiency. H1299 cells stably overexpressing WT, 4A, or 4D NIP30 were treated with indicated concentrations of etoposide, cisplatin, 5-FU, doxorubicin for 48 h followed by MTT assays (n = 4). **c** Xenograft tumors were generated by injecting H1299, and H1299 overexpressing NIP30 4A/4D cells into dorsal flanking sites of nude mice. Two weeks later, mice were treated with cisplatin (5 mg/kg), or doxorubicin (3 mg/kg), or 5-Fluorouracil (20 mg/kg), three times per week i.p. for 2 weeks. All experiments were repeated three times. **d** Quantitation of the results in panel **c**. Values are presented as the means ± SEM. *P < 0.05, ***P < 0.001, (one-way ANOVA). P (lane 1, lane 3) = 0.024, P (lane 1, lane 4) = 1.8E-8, P (lane 1, lane 7) = 3.9E-8, P (lane 1, lane 10) = 3.6E-4, P (lane 4, lane 6) = 0.03, P (lane 7, lane 9) = 0.027, P (lane 10, lane 12) = 1.5E-5. **e** A model depicting the CDC25A–NIP30-REGγ pathway in regulation of the cell cycle and DNA damage response. Source data are provided in a Source Data file.

accumulates in the S phase and is degraded after DNA damage. The physiological significance of the CDC25A–NIP30-REGγ regulatory pathway is supported in vivo by DNA damage experiments and xenograft tumor models, as shown in this work.

Although sequence analysis reveals no known functional domains, homology analysis indicates that NIP30 is evolutionarily conserved, suggesting it is a functionally important protein in mammals. We do not know if NIP30 has additional functions other than regulation of the REGγ proteasome. STRING database and BioGRID database analysis have indicated that NIP30 may have association with some proteasome subunits, such as Psmg2, Psmag3, Psem3[29,41]. However, we do not know if NIP30 may affect degradation of other REGγ substrates. Still, we propose the concept of NIP30 as a putative tumor suppressor, and anticipate upcoming studies with clinical tumor samples will elucidate its role in tumor progression.

CDC25 family members are dual-specificity phosphatases including CDC25A, -B and -C that function to dephosphorylate-specific tyrosine/threonine residues on CDKs[42]. Although we have only demonstrated that CDC25A binds with NIP30 and dephosphorylates NIP30 at serine site 228 and 230 due to lack of phosphor-specific antibodies toward S226 and S227, our findings suggest CDC25A also dephosphorylates the other two serine sites. As an oncogenic protein, CDC25A is overexpressed in many types of cancer and correlates with poor prognosis[43,44]. Consistent with this notion, we have demonstrated CDC25A is a negative regulator of NIP30. We believe this may explain in part why REGγ activity is high in multiple tumors.

It's been an open question whether REGγ function is regulated during the cell cycle, since it regulates cell-cycle regulators, including p21, p16, and p19. We demonstrate here the role of CDC25A allows cell-cycle-dependent function of REGγ. Also, we have linked regulation of REGγ function to DNA damage response by degradation of CDC25A. This provides evidence that CDC25A is involved in regulation of p21 protein levels, independent of p53, in response to DNA damage. Such regulation opens a door for understanding p53-independent cell growth. In sum, this study defines a pathway deciphering regulation of p21, independent of p53, particularly in response to DNA damage via CDC25A and NIP30, and it provides a route for the development of specific proteasome inhibitors.

## Methods

**Mice.** REGγ$^{-/-}$ mice (C57BL/6 genetic background) were provided by Dr. John J. Monaco at the University of Cincinnati and backcrossed for more than ten generations in our SPF animal center. REGγ$^{+/+}$ P53$^{+/-}$ (C57BL/6) mice were originally from JAX Laboratory. NIP30 KO mice were generated by Cyagen Biosciences (Guangzhou, China) using CRISPR/Cas9 technique. Animals were maintained according to the ethical and scientific standards by "ECNU Multifunctional Platform for Innovation(011)".

**Cell culture.** HEK293T, H1299, HCT116, HeLa, and PC9 were purchased from ATCC (USA). All cells were grown in Dulbecco's modified Eagle's medium (DMEM) or RPMI-1640 medium with 10% fetal bovine serum (FBS) and 100 μg/ml penicillin/streptomycin. HCT116 shN and HCT116 shR cells were constructed previously. H1299 shN, H1299 shR, PC9 shN, and PC9 shR cells were generated by stable integration of a retroviral shRNA vector specific for REGγ or a control vector from OriGene (Rockville, MD). H1299 shN & H1299 shR cells stably expressing NIP30 WT, NIP30 4A, or NIP30 4D were generated in our laboratories. 293T-REGγ knockout cells were generated using TALENs. 293T & 293T KO cells with stable expression of NIP30 (221–235) WT or NIP30 (221–235) 4A or NIP30 (221–235) 4D were also generated for this study. The 293-REGγ inducible cell lines were previously described[1].

**Plasmids, constructs, and expression.** pCDNA3.1-flag-REGγ, pCDNA3.1-flag-Last1, and PSG5-HA-HCV truncations were previously generated[6]. PSG5-HA-NIP30, pCDNA3.1-flag-NIP30, pCDNA3.1–GFP-NIP30, and Plvx-EF1α-IRES-Puro-flag-NIP30 were constructed based on the sequence of the human NIP30 gene; a pair of primers was designed to amplify the complete coding region of human NIP30 gene. PCR amplifications were conducted in a final volume of 50 μl

with 2 μl of cDNA from 293T cells, 25 μl of Premix Taq, and 1 μl of each primer (10 μM). PCDNA3-flag-CDC25A was also constructed based on the sequence of human CDC25A in 293T cells. NIP30 siRNA 305# (GUCUGAGGCAGAACUAG AUTT), NIP30 siRNA 407# (CCCUCG-AUCUCUAUAUGAATT) and CDC25A siRNA (CCUGACCGUCACUAUGGACU-U) were synthesized by Genepharma.

**Antibodies and reagents.** Anti-Flag-mouse and β-actin-mouse were obtained from MBL, anti-HA-mouse, anti-γH2AX-rabbit and anti-REGγ-rabbit were obtained from Abmart, anti-NIP30-rabbit, anti-p21-rabbit, and anti-CDC25A-rabbit were obtained from Proteintech. The rabbit antiserum against NIP30 phosphorylated at Ser-228 and Ser 230 were raised by HuaBio Ltd (China) using the synthetic peptide (CSDESS (pS) DSEGTIN / CSDESSSD (pS) EGTIN) as antigen. The antisera were pre-cleaned by affinity chromatography using the corresponding non-phosphorylated peptide (CSDESSSDSEGTIN) coupled to SulfoLink Resin (Thermo Fishfer Scientific), and then purified by affinity chromatography with a phosphor-peptide. NSC95397, a CDC25A inhibitor, was purchased from Sigma. Purified CDC25A (ab90763) protein was purchased from Abcam.

**Western blotting.** Proteins were extracted and processed from cells or tissues as described[45]. Equivalent amounts of total protein from each sample were loaded. After transferring to the nitrocellulose membrane, immunoblots were analyzed using the primary antibodies overnight at 4 °C, then incubated 1–2 h with a fluorescent-labeled secondary antibodies (Jackson ImmunoResearch) and visualized by a LI-COR Odyssey Infrared Imaging System.

**In vitro proteolytic analysis.** NIP30 WT, NIP30 4A, NIP30 4D, and REGγ heptamers protein were purified from bacterial as described[1]. The target protein p21 was translated by an in vitro translation kit (TNT Quick Master Mix 40 μl, Methionine (1 mM) 1 μl, PSG5-T7 promoter-p21 plasmid 2 μg, and add nuclease-free water to 5 μl) as instructed by the producer in 50 μl reactions (Promega, USA). The p21 translation product was aliquated and stored at −80 °C. To quantify the amount of in vitro translated p21 in each 5 μl, we expressed a 2XHis-tagged p21 construct in E. coli BL21 followed by Nickel NTA-affinity chromatography as instructed (Biyotime, China). Commercially available BSA was used to generate a standard curve and quantify the purified His-p21. A series of diluted His-p21 with indicated concentrations (ranged from 5 ng to 50 ng) and 5 μl of the p21 translation product from each reaction were analyzed by WB to estimate the concentration of translated p21 (showing an average of ~15 ng p21 in each 5 μl mix by three experiments). For repeating experiments, p21 from a single translation assay was divided into each tube as indicated in figures. Degradation in vitro was excited by mixing purified REGγ (1 μg), 20S core proteins (0.25 μg), NIP30 WT (1 μg), NIP30 4 A (1 μg), NIP30 4D (1 μg), and p21 (5 μl) to incubate at 30 °C for 30 min in the degradation buffer (20 mM Tris-HCl, 10 mM KCl, 5% glycerol, pH 7.5) in 50 μl of reaction volume. Each mix (combining different proteins) was incubated in parallel, at the same time, for each of the experiments. Decay of p21 is estimated by WB.

**Immunoprecipitation.** Cells were transfected with constructs or treated as explained in the figures. Cells were then scraped into ice-cold PBS and lysed with lysis buffer (50 mM Tris-HCl pH 7.5, 5 mM EDTA, 150 mM NaCl, 1% TritonX-100, 1 mM Na$_3$VO$_4$, 5 mM NaF and protease inhibitors). Specific proteins were immunoprecipitated, followed by three washes with buffer (50 mM Tris-HCl pH 7.5, 5 mM EDTA, 150 mM NaCl, 1 mM Na$_3$VO4, 5 mM NaF and protease inhibitors). The pellet was then suspended in SDS sample buffer for western blot analysis.

**Immunostaining.** Cells were seeded on coverslips in 24-well plates, then washed in cold PBS three times, fixed with 4% paraformaldehyde, and immunostained for NIP30 or REGγ or GFP, as well as DNA staining with 4, 6-diamidino-2-penylindole (DAPI). Then Alexa Fluor 546 (red) goat anti-rabbit antibody (Molecular Probes, OR) was added. Immunofluorescence was visualized by Fluorescence microscopy (Leica).

**Yeast two-hybrid analysis.** The full-length human REGγ cDNA fragment was inserted in frame into the Gal4 DNA-binding domain (DBD) vector pGBKT7 and NIP30 cDNA was cloned in vector pGAD. Detailed methods were performed as described[46].

**Reverse transcriptase–PCR.** The total RNA extracted from cells was followed by treatment with TRIZOL (TakaRa), chloroform, isopropanol, and ethanol. In all, 2 μg of the total RNA was reverse-transcribed in a total volume of 20 μl. For quantitative RT-PCR analysis, reverse-transcribed cDNA was subjected to RT-PCR with Mx3005P (Stratagene). Each experiment was repeated three times. The primers used for quantitative PCR were as follows: for the human version: p21 (5′-GGCAGACCAG CATGACAGATT-3′ and 5′-GCGGATTAGGGCTTCCTC T-3′); for the mouse version: p21 (5′-CCTGGTGATGTCCGACCTG-3′ and 5′-CCATG AGCGCATCGCAATC-3′).

**MTT assay**. In total, $2.5 \times 10^3$ logarithmic-phase cells were seeded per well in 96-well plates and cultured for 24 h, then incubated with 0.5 mg/ml MTT for 4 h and add DMSO for 15 min. Absorbance (490 nm) was measured and analyzed as described[7].

**Phosphatase library screening**. The Human Phosphatase cDNA Expression Library that includes 41 plasimds was donated by Dr. Xinhua Feng at Zhejiang University. Each candidate clone (2 µg) was labeled with numbers for "double-blinded" screening and transiently transfected into 293T cells followed by western blot analyses to determine potential effect on the phosphorylation of NIP30 at 228 site. Clones leading to reduced expression of p-NIP30 were selected for repeated experiments. A clone with dramatic and consistent effects on p-NIP30 after three repeating experiments was sequence verified as CDC25A and proceeded for in vitro dephosphorylation study.

**In vitro dephosphorylation assay**. Immunoprecipitated Flag-NIP30 was eluted with Flag peptides and incubated with 1 µg of bacterially purified CDC25A protein in dephosphorylation reaction buffer (20 mM Tris-HCL (pH 8.5), 75 mM NaCl, 0.57 mM EDTA, 0.033% BSA, and 1 mM DTT) for up to 30 min at 30 °C.

**Isolation of mouse embryo fibroblasts**. In total, 13–14-day embryos were harvested from female NIP30 KO mice and soaked in 70% ethanol. Subsequent steps were described[47].

**Xenograft tumorigenicity analysis**. H1299 stably overexpressing NIP30 WT, NIP30 4A, or NIP30 4D shN and shR cells were prepared for injection. Six-week-old BALB/c male nude mice were used. Cells were subcutaneously injected into the dorsal flanking sites of nude mice at $\times 10^6$ cells in 100 µl per spot (3–6 mice per group). Tumor sizes were measured at 28th day after injection using Vernier calipers, and the final volume was calculated using the following formula. Tumor volume = 1/2 (length × width²). For anticancer drug treatment, tumor-bearing mice (with tumor size of $80\text{–}100^3$ mm) were treated with individual chemotherapeutic agents, as described in figure legends.

**Cell-cycle analyses**. The cells were digested with trypsin, then washed with PBS and fixed with 70% ethanol for 24 h. Cells were stained using PI and measured with BD fascaria flow cytometer in flow cytometry at the Core Facility of ECNU.

**Mass spectrometric analysis-LC–MS/MS and database searching**. Liquid chromatography and tandem mass spectrometry (LC–MS/MS) was carried out using an Orbitrap Fusion Lumos MS (ThermoFisher Scientific) coupled on-line with an Ultimate 3000 HPLC system (ThermoFisher Scientific). MS1 scans were measured with a scan range from 375 to 1500 m/z, resolution set to 120,000, and the AGC target set to $4 \times 105$. MS1 acquisition was performed in top speed mode with a cycle time of 5 s. For MS2 scans, the resolution was set to 30,000, the AGC target 5e4, the precursor isolation width was 1.6 m/z, and the maximum injection time was 100 ms for CID. The CID-MS2 normalized collision energy was 25%.

To identify proteins through database search, we extracted the monoisotopic mass, precursor ion charge state, and ionic strength of the precursor ion and corresponding fragment ion from the LC–MS/MS spectrum according to the Raw_Extract script of Xcalibur v2.4[48], using the Batch-Tag in the development version of Protein Prospector (v 5.10.0) to search the bait database, which contains the ordinary Swissprot database and its random version (SwissProt.2014.12.4. random.concat, a total of 20,196 proteins). Homosapiens was chosen as the species. The mass accuracy of the precursor ion and fragment ion were set to ±20 ppm and 0.6 Da, respectively. Set trypsin as an enzyme and allowed up to one missed cut. The protein N-terminal acetylation, methionine oxidation, glutamine N-terminal conversion to pyroglutamate, and phosphorylation of serine and threonine were selected as variable modifications. The false positive rate of the identified proteins was ≤0.5%.

**Statistical analysis**. Quantitative data were obtained by Prism software (Graph-Pad Software) and displayed as mean ± SEM of independent samples. Statistical analysis of values was performed using one-way analysis of variance (one-way ANOVA or two-tailed $t$ test).

**Ethical statement**. We have complied with all relevant ethical regulations for animal testing and research. The research protocol was approved by the Animal Experiment Ethics Committee of East China Normal University.

**Reporting summary**. Further information on research design is available in the Nature Research Life Sciences Reporting Summary linked to this article.

## Data availability

All original data are available upon request. All the other data supporting the findings of this study are available in the article and Supplementary Information files. Source data are provided with this paper.

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

## Acknowledgements
This work was supported by the National Natural Science Foundation of China (31670882, 31730017, 81672883, 81903244), the Science and Technology Commission of Shanghai Municipality (16ZR1410000, 16QA1401500), and National Institutes of Health grant R01GM074830 to L.H. The Foundation of Guangdong Second Provincial General Hospital (2017-001) and Doctoral workstation foundation of Guangdong Second Provincial General hospital (2019BSGZ008) and Guangzhou Science and Technology Plan Project (202002030404) to LS. We thank ECNU public platform for innovation (011) for mouse facility work. We apprecite Dr. Yongyan Dang for helpful suggestions.

## Author contributions
X.L., L.L., B.H.Z., J.R.X., and S.L. designed and guided the research. X.G., Q.W.W., Y.W., J.L., X.L.Z., Li.P., and L.Z. conducted experiments. H.C., D.W., and S.H.S. raised mice and did genotyping. J.W.C. and G.C. contributed to statistical analysis. Y.Y.C., L.N.P., Z.P.W., and Y.X. supported for transfection studies. J.L., L.H., J.Q., F.F., C.G.W, S. K., and P.Z. provided important research materials and some supplementary results for this project. X.L., X.G., Q.W.W., L.L., R.E.M., D.M.L., and B.W.O. wrote the paper.

## Competing interests
The authors declare no competing interests.
