## [Peer Review File · Nature Communications]

Reviewers' comments:

Reviewer #1 (Remarks to the Author); expert in proteasome regulation:

This is an interesting and potentially important report that describes regulation of REG gamma function via the action of two proteins: NIP30 and CDC25. The authors present data showing that NIP30 directly binds to REG gamma and decrease levels of p21, presumably by enhancing REG gamma activation of proteasome activity. They further show that NIP30 is phosphorylated and that phosphorylation attenuates NIP30/ REG gamma activated p21 degradation. They identify CDC25 as a NIP30 phosphatase and provide evidence that this activity may regulate NIP30/ REG gamma - mediated levels of p21. In general, the data are internally consistent and offer a model for proteasome-dependent control of p21 levels and consequent effects on cell growth.

A main advance of this manuscript is the identification of CDC25 as a regulator of the NIP30/ REG gamma/proteasome axis and the data that support this model. However, many aspects of the presentation are premature, descriptive and problematic.

1. The first section of the results (Figures 1 and 2) describes identification of NIP30 as a REG gamma interacting protein and show that the C-terminus of NIP has a conserved site that is phosphorylated and required for the REG gamma interaction. Similar analysis and conclusions were obtained by Jonik-Nowak et al (ref 32) but are not described in an appropriate context here. This is somewhat misleading. Describing those results more completely in the set-up to the Results section would allow the authors to streamline the paper, focus on what is new, and perhaps go deeper into the mechanisms of the effects reported here.
2. The role of NIP30 phosphorylation is a key mechanistic part of the manuscript. The authors' data are generally consistent with this but there needs to be more direct evidence. I did not find a description of the production of phospho-specific antibodies used in the study – a complete description of the production and validation of these antibodies is essential. The use of these antibodies is introduced in experiments of Figure 3, but should be used earlier. It would be useful to have supporting evidence in the form of direct demonstration of phosphorylation with either mass-spec data, or radioactive phosphate incorporation. There must be reference to (32) where NIP30 phosphorylation of NIP30 by CKII was demonstrated directly.
3. The data in Figure 3 are not very completing. The discrimination of co-localization on the scale used here is not fine enough to demonstrate physical interactions between proteins or assessments of the strengths of these interactions, as claimed in the text. Also, there are many proteins whose highest level of expression occurs in the testis – these results are correlative and not convincing for the claims.
4. As the authors note, the molecular mechanisms by which REG gamma might mediate ubiquitin-independent protein breakdown are not very well understood. In my opinion, this is one of the major problems in understanding how REG gamma functions in the degradation of intact proteins. Although this manuscript fills in some of this understanding, the results still lack mechanistic detail. Steady-state levels of p21 may indeed reflect degradation, but that is not shown directly. As noted above, the data are consistent with the authors' models, but it is possible and likely that NIP30/ REG gamma could affect p21 levels indirectly by other mechanisms. In this regard, the authors should discuss the results of ref 32 that offer either conflicting or alternative possibilities and at least show the complexity of the system. Curiously, others showed that NIP30 inhibited REG gamma proteasome activation (although phos-NIP30 increased the degree of inhibition). In any case, the authors seem to be in a position to test a direct mechanism with a reconstituted in vitro systems. They either have or should be able to prepare pure proteins involved in the proposed regulatory axis and directly test the degradation of non-ubiquitinated p21 by it. As noted early in the Introduction, but then ignored for the rest of the paper, there is controversy about the relative roles of ubiquitin-dependent and ubiquitin-independent degradation of p21. REG gamma could be part of a hybrid proteasome that degrades p21 in an ubiquitin dependent mechanism. The suggested in vitro constitution experiments would help to sort this out.
5. The entire manuscript is in need of extensive editing to correct typographical errors and errors

in English usage and grammar that include the lack of necessary articles, incorrect subject-verb agreement, fractured sentences, and odd or inappropriate word choices. The bibliography contains many incomplete or incorrectly formatted citations. There are too many instances of each of these issues to cite individually here, but in total they make the manuscript distracting and annoying to read. I am sensitive to and appreciate English-as-a-second-language challenges, but it is difficult to believe that all of the authors read and approved this version of the manuscript or that other editing services are not available at Baylor.

Reviewer #2 (Remarks to the Author); expert in cell cycle, DNA damage:

The authors build on the work of another group (Beata Jonik-Nowak et al, Proc Natl Acad Sci, 2018, 115:E6477; PMC6048556) who showed that PIP30/FAM192A (here called NIP30) is phosphorylated by CKII, and that phosphorylated PIP30/NIP30 binds and regulates the function of REGgamma. Here Gao et al confirm those findings and further:

- 1) Show that overexpression of wild-type and phosphomimetic (4A) PIP30/NIP30 posttranscriptionally affects the levels of p21, which is a substrate of REGgamma.
- 2) Show that overexpression of nonphosphorylatable (4D) PIP30/NIP30 reverses the antiproliferative effects of REGgamma knockout
- 3) Claim that CDC25A dephosphorylates PIP30/NIP30, which in turn regulates p21 levels in a p53-independent manner.
- 4) Identify that PIP30/NIP30 is altered in a low percentage of tumors
- 5) Show that overexpression of wild-type and phosphomimetic PIP30/NIP30 increases tumor sensitivity to etoposide, cisplatin, 5-fluorouracil, and doxorubicin
- 6) Claim that PIP30/NIP30 is a tumor suppressor
- 7) Present a model in which CDC25A regulates NIP30/PIP30, which in turn regulates REGgamma that then regulates p21 levels and cell cycle progression.

This is an interesting study. However, there are several serious concerns.

- 1) Although the results showing that altering CDC25A levels affects PIP30/NIP30 phosphorylation in cells is solid, the data showing that CDC25A directly de-phosphorylates PIP30/NIP30 is very weak (Fig. 3E); this analysis used immunopurified PIP30/NIP30 and immunopurified CDC25A. It is very possible that a contaminating phosphatase is responsible for the in vitro dephosphorylation that is observed.
- 2) It is not clear that CDC25A's effects on p21 levels are driven by CDC25A-mediated dephosphorylation of PIP30/NIP30. They could be through another unknown pathway. Does expression of phosphomimetic PIP30/NIP30 prevent CDC25A-induced changes in p21?
- 3) Most of the experiments rely on PIP30/NIP30 overexpression. This is a serious concern because overexpression studies are prone to artefacts. Additionally, given that the authors propose that PIP30/NIP30 is a tumor suppressor, a serious analysis of the effects of losing PIP30/NIP30 would be important to understanding its functions.
- 4) PIP30/NIP30 mutations found in tumors are very infrequent and the effects mutations that are shown in the supplemental figure are not known. Accordingly, there are no data to support the contention that PIP30/NIP30 is a tumor suppressor.
- 5) Although p21 levels are affected by CDC25A and PIP30/NIP30, it is not clear that changes in p21 levels are actually driving cell cycle changes induced by CDC25A and PIP30/NIP30 manipulation.

Minor comments:

- 1) There are many grammatical errors.
- 2) It is not always clear how many times an experiment was done.

- 3) Abbreviations are not always defined.
- 4) Figures are too small to read, especially annotations and symbols on graphs, even when zoomed in on.
- 5) Some figures are not appropriately labeled. For example, in Fig. 6B, the x-axes are not labeled and the y-axes are labeled as "survival rate." If an MTT assay was used, this is not a survival rate.

Reviewer #3 (Remarks to the Author); expert in p53, mouse models:

The manuscript by Gao, et al. focuses on NIP30 as an inhibitor of the REG γ proteasome, which is oncogenic and distinct from the 26S proteasome. The authors uncovered a tumor suppressor role for NIP30 by its interactions with REG γ . mapping the amino acids of interaction and uncovering a phosphorylation switch for this interaction. CDC25A was identified as the major regulatory phosphatase for this inhibitory switch during cell cycle and in response to DNA damage. Cell cycle inhibitor p21 is a major target of the CDC25A-NIP30- REG γ axis, which induces regulation in a p53-dependent manner. Multiple approaches were employed in these studies: IP/mass spec; yeast two-hybrid; co-IP; domain and amino acid interaction mapping; xenograft tumor analyses of expressing, mutated and KD of key players; screening of a phosphatase library; cell cycle and DNA damage studies and UV-response of the skin taken from newborn REG γ WT, KO, Tpr53 KO and REG γ ;Tpr53 double KO; and chemosensitivity studies with cell lines and xenografts.

Overall, this is a very thorough analysis that pinpoints a specific phosphorylation switch control of proteasome regulation of p21 during cell cycle and DNA damage response. The studies are primarily cell line-based but are also complemented by xenograft, genetically modified mice and TCGA data. This work suggests a new target and approach toward effectively treating p53-dysfunctional cancers with chemotherapeutics, a major problem in the clinic. This finding is likely to be of considerable interest to the community.

The major problem with the manuscript is poor preparation, improper use of English and inexact terminology. The authors should employ an English editor and science writer to resubmit a better manuscript.

Specific examples of inexact terminology include "tight binding", found in the Abstract and within the manuscript. How is "tight binding" measured and what are the values for binding constants that would indicate "tight"? Quantifying this binding is needed for such terms to be used. Showing co-localization in immunofluorescence microscopy is insufficient and non-robust (Suppl. Fig. 3A).

Examples of poor manuscript preparation are the numerous grammatical errors, incorrect word usage that obscures the likely intended meaning (a couple of examples: "difference cells"? "Hiked levels"?), and some references improperly formatted. Additionally, some figure legends lack sufficient detail (Fig. 1A, what is N1 and S1, for example). References are sometimes lacking. A major example of this is the human Ser/Thr phosphatase library screen. There is no reference for this library or any data for the screen shown. These data should be included as supplemental data or referenced.

Response Letter

We have tried our best within limited time to complete a series of experiments suggested by the reviewer. Altogether 40 pieces of new or revised data have been incorporated into the revised manuscript. We believe that this revised manuscript should be qualified for publication in *Nat. Commun.* The following is our point-by-point response to the reviewers' comments.

Arrangement of Figures

Original	In Revision
	Fig. 3E (new)

Supplement Figures

Original	In Revision
	Fig. S2C(new)
Fig S2C	Fig S2D(relocation)
Fig S2D	Fig S2E(relocation)
Fig S2E	Fig S2F(relocation)
Fig S3F	Fig S2G(relocation)
Fig. S3	delete
	Fig. S3D(new)
	Fig. S3E(new)
	Fig. S3F(new)
Fig. S4A	Fig.S3A(relocation)
Fig. S4B	Fig. S3B(relocation)
Fig. S4C	Fig. S3C(relocation)
Fig. S4D	Fig. S3G(relocation)
Fig. S4E	Fig.S3H(relocation)
Fig. S4F	Fig.S3I(relocation)
Fig.S5	Fig.S4(relocation)
Fig.S6A	Fig.S5A(relocation)
Fig.S6B	Fig.S5B(relocation)
Fig.S6C	Fig.S5C(relocation)
Fig.S6D	Fig.S5D(relocation)

	Fig.S5E (new)
	Fig.S5F (new)
	Fig.S5G(new)
	Fig.S5H (new)
	Fig.S5I(new)
	Fig.S5J(new)
	Fig.S5K(new)
Fig.3E	Fig.S5L(relocation)
Fig.S7A	Fig.S6A(relocation)
Fig.S7B	Fig.S6B(relocation)
Fig.S7C	Fig.S6C(relocation)
	Fig.S6D(new)
	Fig.S6E(new)
	Fig.S6F(new)
Fig.S8	Fig.S7
Fig.S8G	deletion
Fig.S8H	deletion
	Fig.S7G(new)
	Fig.S7H(new)

Reviewers' comments:

Reviewer #1 (Remarks to the Author); expert in proteasome regulation:

This is an interesting and potentially important report that describes regulation of REG gamma function via the action of two proteins: NIP30 and CDC25. The authors present data showing that NIP30 directly binds to REG gamma and decrease levels of p21, presumably by enhancing REG γ activation of proteasome activity. They further show that NIP30 is phosphorylated and that phosphorylation attenuates NIP30/REG γ activated p21 degradation. They identify CDC25 as a NIP30 phosphatase and provide evidence that this activity may regulate NIP30/REG γ - mediated levels of p21. In general, the data are internally consistent and offer a model for proteasome-dependent control of p21 levels and consequent effects on cell growth.

A main advance of this manuscript is the identification of CDC25 as a regulator of the NIP30/ REG γ /proteasome axis and the data that support this model. However, many aspects of the presentation are premature, descriptive and problematic.

1. The first section of the results (Figures 1 and 2) describes identification of NIP30 as a REG γ interacting protein and show that the C-terminus of NIP has a conserved site that is phosphorylated and required for the REG γ interaction. Similar analysis and conclusions were obtained by Jonik-Nowak et al (ref 32) but are not described in an appropriate context here. This is somewhat misleading. Describing those results more completely in the set-up to the Results section would allow the authors to streamline the paper, focus on what is new, and perhaps go deeper into the mechanisms of the effects reported here.

Answer: We have described similar observations by Jonik-Nowak et al (ref 32) in the results section, particularly NIP30-REG γ interaction, Casein kinase II involved in NIP30 phosphorylation, and potential mechanistic mode in discussion.

2. The role of NIP30 phosphorylation is a key mechanistic part of the manuscript. The authors' data are generally consistent with this but there needs to be more direct evidence. I did not find a description of the production of phospho-specific antibodies used in the study – a complete description of the production and validation of these antibodies is essential. The use of these antibodies is introduced in experiments of Figure 3, but should be used earlier. It would be useful to have supporting evidence in the form of direct demonstration of phosphorylation with either mass-spec data, or radioactive phosphate incorporation. There must be reference to (32) where NIP30 phosphorylation of NIP30 by CKII was demonstrated directly.

Answer: Thank you for comments. We have included description of the phospho-antibody production in Materials and Methods.

We have also provided the mass-spec data of NIP30 phosphorylation in Supplemental data 3D and 3E, showing both NIP30 S228 and S230 sites are phosphorylated. Ref 32

is cited to indicate CKII is a major kinase required for NIP30 phosphorylation.

3. The data in Supple Figure 3 are not very completing. The discrimination of co-localization on the scale used here is not fine enough to demonstrate physical interactions between proteins or assessments of the strengths of these interactions, as claimed in the text. Also, there are many proteins whose highest level of expression occurs in the testis – these results are correlative and not convincing for the claims.

Answer: We agree with the comments. Given the issues in the original Supple Fig 3 and we have already had enough evidence to show NIP30-REG γ interaction (in addition to published reference), we have deleted the Figure S3. This will not have any impact on our major conclusions.

4. As the authors note, the molecular mechanisms by which REG γ might mediate ubiquitin-independent protein breakdown are not very well understood. In my opinion, this is one of the major problems in understanding how REG γ functions in the degradation of intact proteins. Although this manuscript fills in some of this understanding, the results still lack mechanistic detail. Steady-state levels of p21 may indeed reflect degradation, but that is not shown directly. As noted above, the data are consistent with the authors' models, but it is possible and likely that NIP30/REG γ could affect p21 levels indirectly by other mechanisms. In this regard, the authors should discuss the results of ref 32 that offer either conflicting or alternative possibilities and at least show the complexity of the system. Curiously, others showed that NIP30 inhibited REG γ proteasome activation (although phos-NIP30 increased the degree of inhibition). In any case, the authors seem to be in a position to test a direct mechanism with a reconstituted *in vitro* systems. They either have or should be able to prepare pure proteins involved in the proposed regulatory axis and directly test the degradation of non-ubiquitinated p21 by it. As noted early in the Introduction, but then ignored for the rest of the paper, there is controversy about the relative roles of ubiquitin-dependent and ubiquitin-independent degradation of p21. REG γ could be part of a hybrid proteasome that degrades p21 in an ubiquitin dependent mechanism.

The suggested in vitro constitution experiments would help to sort this out.

Answer: We have followed the suggestion and determined whether NIP30 inhibited the REG γ 's degradation of p21 directly in Supplemental data 3F. We used cell-free proteolysis system to incubate purified NIP30 WT (not phosphorylated in bacteria), NIP30 4A or NIP30 4D with a combination of REG γ and 20S proteasome. The results exhibit that the NIP30 4D (phosphorylation-mimetic), but not the WT or NIP30 4A, blocks p21 degradation by the REG γ -proteasome. Regarding degradation of non-ubiquitinated p21, James Roberts' lab has demonstrated ub-independent degradation of p21 beautifully (Molecular Cell, 2000, 2004, 2017).

We noticed the finding in ref 32 that PIP30 differentially alters degradation of a panel of standard proteasome peptide substrates by the PA28 γ -activated proteasome, reflecting a substrate specificity issue or activator spectrum potential. We discussed this briefly in the DISCUSSION. With our continued efforts, we should be able to address all the questions in the future.

5. The entire manuscript is in need of extensive editing to correct typographical errors and errors in English usage and grammar that include the lack of necessary articles, incorrect subject-verb agreement, fractured sentences, and odd or inappropriate word choices. The bibliography contains many incomplete or incorrectly formatted citations. There are too many instances of each of these issues to cite individually here, but in total they make the manuscript distracting and annoying to read. I am sensitive to and appreciate English-as-a-second-language challenges, but it is difficult to believe that all of the authors read and approved this version of the manuscript or that other editing services are not available at Baylor.

Answer: The revised manuscript has been edited by Dr. Robb Moses and proofed by Dr. Bert O'Malley..

Reviewer #2 (Remarks to the Author); expert in cell cycle, DNA damage:

The authors build on the work of another group (Beata Jonik-Nowak et al, Proc Natl Acad Sci, 2018, 115:E6477; PMC6048556) who showed that PIP30/FAM192A (here called NIP30) is phosphorylated by CKII, and that phosphorylated PIP30/NIP30 binds and regulates the function of REGgamma. Here Gao et al confirm those findings and further:

- 1) Show that overexpression of wild-type and phosphomimetic (4A) PIP30/NIP30 posttranscriptionally affects the levels of p21, which is a substrate of REGgamma.
- 2) Show that overexpression of nonphosphorylatable (4D) PIP30/NIP30 reverses the antiproliferative effects of REGgamma knockout
- 3) Claim that CDC25A dephosphorylates PIP30/NIP30, which in turn regulates p21 levels in a p53-independent manner.
- 4) Identify that PIP30/NIP30 is altered in a low percentage of tumors
- 5) Show that overexpression of wild-type and phosphomimetic PIP30/NIP30 increases tumor sensitivity to etoposide, cisplatin, 5-fluorouracil, and doxorubicin
- 6) Claim that PIP30/NIP30 is a tumor suppressor
- 7) Present a model in which CDC25A regulates NIP30/PIP30, which in turn regulates REGgamma that then regulates p21 levels and cell cycle progression.

This is an interesting study. However, there are several serious concerns.

1) Although the results showing that altering CDC25A levels affects PIP30/NIP30 phosphorylation in cells is solid, the data showing that CDC25A directly de-phosphorylates PIP30/NIP30 is very weak (Fig. 3E); this analysis used immunopurified PIP30/NIP30 and immunopurified CDC25A. It is very possible that a contaminating phosphatase is responsible for the in vitro dephosphorylation that is observed.

Answer: Good comments/suggestions! To exclude possible contamination by cellular phosphatases, we incubated bacterially purified CDC25A protein with immunoprecipitated NIP30. The results showed a decreased phosphorylation of NIP30 S228 and NIP30 S230 in Figure 3E (as a control, the lane labeled “0” was

also incubated for 30 min at 30 °C).

2) It is not clear that CDC25A's effects on p21 levels are driven by CDC25A-mediated dephosphorylation of PIP30/NIP30. They could be through another unknown pathway. Does expression of phosphomimetic PIP30/NIP30 prevent CDC25A-induced changes in p21?

Answer: We followed your suggestion and performed overexpression of CDC25A in normal cells and cells stably overexpressing phosphor-mimetic or phosphor-defective NIP30. We conclude that expression of phosphor-mimetic PIP30/NIP30 prevents CDC25A-induced changes in p21 (Supplemental data 5F).

3) Most of the experiments rely on PIP30/NIP30 overexpression. This is a serious concern because overexpression studies are prone to artefacts. Additionally, given that the authors propose that PIP30/NIP30 is a tumor suppressor, a serious analysis of the effects of losing PIP30/NIP30 would be important to understanding its functions.

Answer: Good comments/suggestions! We employed two siRNA specific for NIP30 to determine the impact of NIP30 deficiency on p21 degradation in Supplemental data 2C. MEF cells isolated from NIP30 KO mice were used to substantiate CDC25A function in Supplemental data 5I.

4) PIP30/NIP30 mutations found in tumors are very infrequent and the effects mutations that are shown in the supplemental figure are not known. Accordingly, there are no data to support the contention that PIP30/NIP30 is a tumor suppressor.

Answer: We have tuned down the description about the frequency in NIP30 mutation. Since we do not have real *in vivo* animal model such NIP30 4A/4D knock-in mice, we also changed our statement as putative tumor suppressor or suppressive functions about NIP30.

5) Although p21 levels are affected by CDC25A and PIP30/NIP30, it is not clear that

changes in p21 levels are actually driving cell cycle changes induced by CDC25A and PIP30/NIP30 manipulation.

Answer: Good comments/suggestions! To determine if p21 levels actually drive cell cycle changes following CDC25A/NIP30 manipulation, DNA content was analyzed by flow cytometry in Normal, NIP30 4A/4D overexpressing, NIP30 knockdown, or CDC25A manipulated 293T /H1299 cells (Supplemental data 6D, 6E, 6F). The DNA content in G1 phase was affected with a corresponding change in S phase upon manipulation of NIP30 or CDC25A.

Minor comments:

- 1) There are many grammatical errors.
- 2) It is not always clear how many times an experiment was done.
- 3) Abbreviations are not always defined.
- 4) Figures are too small to read, especially annotations and symbols on graphs, even when zoomed in on.
- 5) Some figures are not appropriately labeled. For example, in Fig. 6B, the x-axes are not labeled and the y-axes are labeled as “survival rate.” If an MTT assay was used, this is not a survival rate.

Answer: all these issues were taken care of.

Reviewer #3 (Remarks to the Author); expert in p53, mouse models:

The manuscript by Gao, et al. focuses on NIP30 as an inhibitor of the REG γ proteasome, which is oncogenic and distinct from the 26S proteasome. The authors uncovered a tumor suppressor role for NIP30 by its interactions with REG γ . mapping the amino acids of interaction and uncovering a phosphorylation switch for this interaction. CDC25A was identified as the major regulatory phosphatase for this inhibitory switch during cell cycle and in response to DNA damage. Cell cycle inhibitor p21 is a major target of the CDC25A-NIP30- REG γ

axis, which induces regulation in a p53-dependent manner. Multiple approaches were employed in these studies: IP/mass spec; yeast two-hybrid; co-IP; domain and amino acid interaction mapping; xenograft tumor analyses of expressing, mutated and KD of key players; screening of a phosphatase library; cell cycle and DNA damage studies and UV-response of the skin taken from newborn REG γ WT, KO, Tpr53 KO and REG γ ;Tpr53 double KO; and chemosensitivity studies with cell lines and xenografts.

Overall, this is a very thorough analysis that pinpoints a specific phosphorylation switch control of proteasome regulation of p21 during cell cycle and DNA damage response.

Examples of poor manuscript preparation are the numerous grammatical errors, incorrect word usage that obscures the likely intended meaning (a couple of examples: “difference cells”? “Hiked levels”?), and some references improperly formatted. Additionally, some figure legends lack sufficient detail (Fig. 1A, what is N1 and S1, for example). References are sometimes lacking. A major example of this is the human Ser/Thr phosphatase library screen. There is no reference for this library or any data for the screen shown. These data should be included as supplemental data or referenced.

Answer: all issues regarding manuscript preparation are taken care of. The initial screen of phosphatase library was performed in a double-blinded way. Therefore, the representative Western blot result included in Supplemental data 5E just showed numbers. Promising candidate clones were further validated by repeating experiments. The clone with reproducible effects on pNIP30 was verified by sequencing analysis (Supplemental data 5F) and blast in NCBI database (Supplemental data 5G). A reference has been included to declare the source of the library. Experimental procedures were described in M&M.

Reviewers' comments:

Reviewer #1 (Remarks to the Author):

This is a revised manuscript that examines the relative roles of CDC25, NIP30, and REGγ on proteasome-dependent degradation of p21 and consequent cell growth. As with the previous version of the manuscript, the results allow the authors to present a model whereby NIP30 regulates REGγ activated proteasome activity via CDC25 control of NIP30 phosphorylation status. This is an interesting and potentially important set of results.

The authors have responded to my original concerns to improve the manuscript. However, in my opinion, their responses are not completely satisfactory and in most cases consist of minor additions to a largely unchanged text. For example, I thought that reference to previous work by others was not sufficiently acknowledged in the setup to this work and thought that a satisfactory response would include a more serious reworking of the introduction and first part of the results. Instead the authors have simply added references in several places.

As noted in my original comments, I continue to think that a major unresolved issue here is the molecular mechanism by which REGγ/NIP30 regulates proteasome action against protein substrates such as p21 and suggested an *in vitro* experiment that could directly examine this process and test the authors' models. The authors have provided such an experiment (Supplemental Figure 3F), but it is incompletely described and therefore difficult to evaluate; there is no indication of relative concentrations of proteins in the assay, details of the incubation conditions, validation that the single time point is an appropriate measure of rate, etc. The use of *in vitro* translated (in what system?) p21 (instead of bacterially expressed and purified p21) introduces lysates into the system. Maybe it makes no difference, but who knows? The single p21-only "control" is not sufficient without a detailed description of the assay conditions and components.

The model figure (6E) is somewhat confusing and perhaps misleading since there is no direct experimental evidence to show whether the effect of CDC25 action on NIP30 phosphorylation affects proteasome activity by blocking REGγ/NIP30 interaction in isolation or whether this occurs within the proteasome complex. This is an issue that could add considerable mechanistic detail to the work and could be tested directly, but is not done so here.

Blanket statements in many figure legends, such as "All experiments were repeated three times." are not entirely clear. Does that mean the complete xenograph experiment shown in Figure 6C and 6D was done independently three times? If so, why not combine all of the data?

The presentation is improved but there are still many examples of grammatical errors and odd syntax throughout.

Reviewer #2 (Remarks to the Author):

My major concerns have been addressed. However, although the grammar and word usage have been much improved, the text is still rough. Also, the figures are still very small and very difficult to read.

Reviewer #3 (Remarks to the Author):

The resubmitted manuscript is much improved and addresses the majority of problems in the first submission. The major claims are substantiated and present new evidence regarding tumor

suppressor and p53-independent functions.

A few problems with wording remain in the Discussion: lines 495, 498, 527 and 542.

Overall, a much improved submission.

Thank you for the rapid and helpful review of our revised manuscript, NCOMMS-19-12459A. We are pleased that Reviewers #2 and #3 are satisfied, We have responded to the comments of Reviewer #1 as follows:

1. The authors have responded to my original concerns to improve the manuscript. However, in my opinion, their responses are not completely satisfactory and in most cases consist of minor additions to a largely unchanged text. For example, I thought that reference to previous work by others was not sufficiently acknowledged in the setup to this work and thought that a satisfactory response would include a more serious reworking of the introduction and first part of the results. Instead the authors have simply added references in several places.

Answer: We have included the major conclusion of the work done by Jonik-Nowak et al. in the introduction, citing the work in introduction, results and discussion. We feel this gives our work better perspective.

2. As noted in my original comments, I continue to think that a major unresolved issue here is the molecular mechanism by which REG gamma/NIP30 regulates proteasome action against protein substrates such as p21 and suggested an in vitro experiment that could directly examine this process and test the authors' models. The authors have provided such an experiment (Supplemental Figure 3F), but it is incompletely described and therefore difficult to evaluate; there is no indication of relative concentrations of proteins in the assay, details of the incubation conditions, validation that the single time point is an appropriate measure of rate, etc. The use of in vitro translated (in what system?) p21 (instead of bacterially expressed and purified p21) introduces lysates into the system. Maybe it makes no difference, but who knows? The single p21-only "control" is not sufficient without a detailed description of the assay conditions and components.

Answer: We have added experimental details in Materials and Methods as we have been following the well-established in vitro degradation system (Li et al. Cell, 2006; Li et al. Mol. Cell, 2007; Xu et al. Nature Communications, 2016) to perform cell-free proteolysis experiments. What we did not show in a previous publication (Mol. Cell 2007) was that purified p21 (either from bacteria or baculovirus) cannot be degraded by the REGgamma-proteasome, suggesting a lack of critical components in the purified system or a required modification of the target protein. Therefore, we did not try purified but used in vitro translated p21. In our revision, we have included purified NIP30 (construct representing either phosphorylation or dephosphorylation status) in the cell-free system. Our results demonstrate that in a reconstituted cell-free system, phosphorylation mimetic NIP30 4D, but not phosphorylation-defective mutant NIP30 4A, can interfere with REGgamma-directed destruction of a substrate. In the newly revised version, we

have quantified the relative abundance of the p21 proteins (Suppl Fig 3G). To clarify, we also have included more detailed information in the Suppl Fig 3F legend and in the text.

3. The model figure (6E) is somewhat confusing and perhaps misleading since there is no direct experimental evidence to show whether the effect of CDC25 action on NIP30 phosphorylation affects proteasome activity by blocking REG γ /NIP30 interaction in isolation or whether this occurs within the proteasome complex. This is an issue that could add considerable mechanistic detail to the work and could be tested directly, but is not done so here.

Answer: Although this is a new question, we are happy to address it. We did not show in this model if CDC25A affects proteasome activity or not by blocking REG γ /NIP30 interaction **in isolation or in the complex**. This is a very good suggestion for us to follow up. To leave the door open, we have left the model as is but raised this point in the text (ln 449). Since NIP30 must be phosphorylated to bind REG γ we assume that dyad formation has occurred, but the figure leaves that open.

4. Blanket statements in many figure legends, such as “All experiments were repeated three times.” are not entirely clear. Does that mean the complete xenograph experiment shown in Figure 6C and 6D was done independently three times? If so, why not combine all of the data?

Answer: Yes, we have repeated the experiments in Fig 6C and all other experiments three times or more.

5. The presentation is improved but there are still many examples of grammatical errors and odd syntax throughout.

Answer: We have reviewed the manuscript with our colleagues and co-authors to improve syntax and usage.

We hope these responses will meet the expectations of the Nature Communications review team and are satisfactory.

Respectfully,

Xiaotao Li

Professor of Biochemistry

REVIEWERS' COMMENTS:

Reviewer #1 (Remarks to the Author):

This manuscript has been revised in response to my previous comments. Some responses are satisfactory, but others, in my opinion, remain unsatisfactory. I continue to believe that acknowledgment of earlier work (ref 32) is inadequate with respect to the findings therein and their relationship to data in the current manuscript (e.g. insight to and identification of the phosphorylation sites, effects of phosphorylation, etc, etc). I also consider the description, conduct and analysis of the in vitro degradation experiment I questioned previously to be inadequate - including weight-amounts of some proteins is a bare minimum of new information. Finally, although somewhat improved, the presentation still includes a number of odd word choices and syntax throughout and there many typos and inappropriate formatting, especially in the Reference section.

Response to Referees

We have responded to the comments of Reviewer #1 as follows:

This manuscript has been revised in response to my previous comments. Some responses are satisfactory, but others, in my opinion, remain unsatisfactory. I continue to believe that acknowledgment of earlier work (ref 32) is inadequate with respect to the findings therein and their relationship to data in the current manuscript (e.g. insight to and identification of the phosphorylation sites, effects of phosphorylation, etc, etc). I also consider the description, conduct and analysis of the in vitro degradation experiment I questioned previously to be inadequate - including weight-amounts of some proteins is a bare minimum of new information. Finally, although somewhat improved, the presentation still includes a number of odd word choices and syntax throughout and there many typos and inappropriate formatting, especially in the Reference section.

Response: We have added the references # 32 in the section regarding “NIP30 interacts with and inhibits REGy”. We further supplemented experimental details for “in vitro degradation” in Methods and Materials.

We hope these responses will meet the expectations of the Nature Communications review team and are satisfactory.